# Altered Regional Activity and Network Homogeneity within the Fronto-Limbic Network at Rest in Medicine-Free Obsessive–Compulsive Disorder

**DOI:** 10.3390/brainsci12070857

**Published:** 2022-06-29

**Authors:** Yunhui Chen, Yangpan Ou, Dan Lv, Zengyan Yu, Tinghuizi Shang, Jidong Ma, Chuang Zhan, Zhenning Ding, Xu Yang, Jian Xiao, Ru Yang, Zhenghai Sun, Guangfeng Zhang, Xiaoping Wang, Wenbin Guo, Ping Li

**Affiliations:** 1Department of Psychiatry, Qiqihar Medical University, Qiqihar 161006, China; chenyh123@sina.com (Y.C.); lvdan_9999@163.com (D.L.); yuzengyan@163.com (Z.Y.); ShangTHZ188@163.com (T.S.); dzn_conquer@163.com (Z.D.); yangxu990629@126.com (X.Y.); tslgxj@126.com (J.X.); zhsun1980@sina.com (Z.S.); 2Department of Psychiatry, National Clinical Research Center for Mental Disorders, The Second Xiangya Hospital of Central South University, Changsha 410011, China; ouyangpan33@csu.edu.cn (Y.O.); xiaop6@gmail.com (X.W.); 3Department of Psychiatry, Baiyupao Psychiatric Hospital of Harbin, Harbin 150026, China; bypywk@163.com (J.M.); zhanchuang1010@163.com (C.Z.); 4Department of Radiology, The Second Xiangya Hospital of Central South University, Changsha 410011, China; yr_smu@126.com; 5Department of Radiology, Shenzhen Third People’s Hospital, Shenzhen 518000 China; QYFSZGF@qmu.edu.cn

**Keywords:** obsessive–compulsive disorder, fractional amplitude of low-frequency fluctuations, network homogeneity, support vector machine, resting state

## Abstract

Functional abnormalities in brain areas within the fronto-limbic network have been widely reported in obsessive–compulsive disorder (OCD). However, region- and network-level brain activities of the fronto-limbic network at rest have not been simultaneously investigated in OCD. In this study, 40 medicine-free and non-comorbidity patients with OCD and 38 age-, education-, and gender-matched healthy controls (HCs) underwent a resting-state functional magnetic-resonance-imaging scan. Fractional amplitude of low-frequency fluctuations (fALFF), network homogeneity (NH), and support vector machine were used to analyze the data. Patients with OCD showed increased fALFF in the right orbital frontal cortex (OFC), increased NH in the left OFC, and decreased NH in the right putamen. Decreased NH of the right putamen was negatively correlated with the Y-BOCS total and compulsive behavior scores. Furthermore, a combination of NH in the left OFC and right putamen could be applied to differentiate OCD from HCs with optimum specificity and sensitivity. The current findings emphasize the crucial role of the fronto-limbic network in the etiology of OCD.

## 1. Introduction

In addition to intrusive thoughts and/or compulsive behavior, anxiety, dysregulated fear, and uncertainty are the primary clinical characteristics of obsessive–compulsive disorder (OCD), which may affect daily life and social function of the patients [1,2,3,4]. Increasing findings demonstrated that these apparent characteristics may be caused by the dysfunction of brain circuits rather than a single brain region [5].

The cortico-striato-thalamo-cortical circuit including cortical areas, striatum, and thalamus has been a considerable model involving the neural basis of OCD over the years [6,7,8]. According to recent reviews, five parallel and segregated neural networks are related to different clinical characteristics of OCD: the fronto-limbic, dorsal cognitive, ventral cognitive, ventral affective, and sensorimotor networks [6]. The fronto-limbic circuit connecting the ventromedial prefrontal cortex (vmPFC), amygdala, and hippocampus is crucial for dysregulated fear and uncertainty of OCD [6].

Altered structure and function of the fronto-limbic network have been discovered in patients with OCD. For example, increased and/or decreased gray-matter volume in the vmPFC, hippocampus, and thalamus has been found in OCD [9,10,11]. Dysfunctional activities during emotional processing and increased activities during uncertainty and decision-making within the fronto-limbic network have been noted in OCD [2,12,13]. At the same time, hyperconnectivities within the fronto-limbic network (i.e., between caudate and orbital frontal cortex [OFC] and anterior cingulate cortex [ACC], and between vmPFC and ACC) have been discovered in OCD at rest [14]. Moreover, decreased amygdala-vmPFC functional connectivity and right amygdala degree centrality may predict a better outcome of cognitive behavior therapy in OCD [15,16]. The above-mentioned research related alterations in the fronto-limbic network to the pathophysiology of OCD.

Previous studies demonstrated that the local and network properties of brain function at rest are closely related [17,18,19]. However, region- and network-level brain activities of the fronto-limbic network at rest have not been simultaneously investigated in OCD. In the current research, fractional amplitude of low-frequency fluctuations (fALFF) combined with network homogeneity (NH) were applied to comprehensively assess the local and network properties of the fronto-limbic network in OCD at rest. FALFF explores the intensity of regional brain spontaneous activities, and it can be used to detect the regional brain activities at rest [20]. The NH approach evaluates the synchronization of a voxel with all other brain voxels in a brain network and supplies an assessment of a given network with an unbiased hypothesis-driven manner [21]. The combination of these two methods may provide complementary information underlying the fronto-limbic network involved in OCD [22]. We hypothesized that OCD could display changed fALFF and/or NH values within the fronto-limbic network at rest, which could be related to clinical characteristics (i.e., symptom severity and illness duration) and can be used to identify patients with OCD from HCs.

## 2. Materials and Methods

### 2.1. Participants

The participants comprised 40 patients with OCD (13 females and 27 males) and 38 HCs (13 females and 25 males). They were all Han Chinese, right-handed, and 18–50 years old. All subjects were informed of the aims and procedures of our study and sighed an informed consent form. This research was confirmed by the Research Ethics Committee of Qiqihar Medical University. 

The diagnoses for each patient with OCD were conducted by two psychiatrists according to the Structured Clinical Interview for DMS-IV (SCID) patient version. The clinical symptoms of OCD were evaluated with the Yale–Brown Obsessive–Compulsive Scale (Y-BOCS), Hamilton Anxiety Rating Scale (HAMA), and 17-item Hamilton Depression Rating Scale (HAMD). All patients had Y-BOCS total score ≥16 and 17-HAMD score <18 and were psychotropic medication-free for at least 4 weeks. Eighteen patients with OCD were drug-naive; fourteen patients had a history of antiobsessive/antidepressant/anxiolytic medication (i.e., selective serotonin reuptake inhibitors, serotonin and norepinephrine reuptake inhibitors, benzodiazepines, and buspirone); and eight patients had a history of antipsychotics (i.e., aripiprazole 5–20 mg/day). None of the patients were undergoing any systematic behavioral therapy or cognitive behavioral therapy. HCs were screened using the SCID non-patient version. Exclusion criteria of all participants included (1) any other psychiatric disorder; (2) serious physical or neurological disease; (3) drug or alcohol dependence; and (4) contraindication for an MRI scan.

### 2.2. MRI Data Acquisition

MRI images were obtained with a 3.0-Tesla GE 750 Signa-HDX scanner. All subjects were instructed to lie quietly, close their eyes, and stay awake. Resting-state functional magnetic resonance imaging (rs-fMRI) data were acquired with an echo-planar imaging sequence: 33 axial slices, 2000 ms repetition time, 30 ms echo time, 3.5 mm slice thickness, 0.6 mm inter-slice gap, 90° flip angle, 200 × 200 mm^2^ field of view, 64 × 64 data matrix, and 240 volumes in total. All subjects manifested no significantly structural abnormalities in the brain. 

### 2.3. fMRI Data Preprocessing

The Data Processing Assistant for Brain Imaging (DPABI) software was used to conduct the fMRI data preprocessing [23], which included the following steps: discarding the first 10 functional volumes, slice timing and head motion correction, normalization and spatial resampling to 3 × 3 × 3 mm^3^, smoothing with an isotropic Gaussian kernel of 8 mm, band-pass filtering (0.01–0.08 Hz), regression of the nuisance covariates (i.e., cerebrospinal fluid, white matter, and 24 motion parameters), and scrubbing of time points with a threshold of 0.2 mm of framewise displacement (FD) [24].

### 2.4. Fronto-Limbic Network Mask Identification

We established the fronto-limbic network mask based on the anatomical automatic labeling templates, including the superior medial frontal gyrus, superior frontal gyrus, middle frontal gyrus, inferior frontal gyrus, ACC, medial OFC, amygdala, thalamus, and hippocampus [25] (Appendix A).

### 2.5. FALFF Analysis

FALFF analysis was conducted with REST software with the following steps [26]. First, the time series were transformed into the frequency domain to obtain the power spectrum using fast Fourier transform. Second, the square root of the spectral power spectrum was calculated at each frequency and averaged across 0.01–0.08 Hz in each voxel. Finally, for standardization purposes, the sum of amplitude was further divided by the whole frequency range within the fronto-limbic network mask for each subject, and the fALFF value was obtained.

### 2.6. NH Analysis

NH analysis was conducted based on MATLAB. For each participant, correlation coefficients were obtained for each voxel in relation to all other voxels within the fronto-limbic network mask. The mean correlation coefficient refers to the NH of a given voxel. Then average NH of each voxel of the fronto-limbic network was generated. A Gaussian kernel of 4 mm was used to smooth the averaged NH maps twice [21]. Finally, the NH maps of the fronto-limbic network were used for group comparison.

### 2.7. Statistical Analysis

Two-sample *t*-tests and chi-square test were used to analyze the continuous variables and categorical data of demographic characteristics between OCD and HCs, respectively. Two-sample *t*-tests were utilized to compare the fALFF and NH values between the two groups in voxel-wise within the fronto-limbic network, with age, sex, and mean FD values as covariates for minimizing the potential effects. *p* < 0.05 corrected by Gaussian random field (voxel significance: *p* < 0.001, cluster significance: *p* < 0.05) was the significant level.

Pearson analysis was conducted to evaluate the correlation between the altered fALFF/NH values (mean *z* values of fALFF/NH) and clinical characteristics in OCD. *p* < 0.05 (Bonferroni corrected) was the significant level.

Support vector machine (SVM) analysis was conducted using LIBSVM (https://github.com/cjlin1/libsvm (accessed on 16 August 2020)) to explore whether changed fALFF and/or NH values within the fronto-limbic network can be utilized to identify OCD from HCs. The SVM model consisted of a training dataset for selecting discriminating clusters and testing dataset for checking the classification performance. SVM analysis was executed via a “leave-one-out” approach. 

## 3. Results

### 3.1. Demographic and Clinical Data 

Age, gender, education, and FD values were not significantly different between the patients with OCD and HCs. However, patients with OCD had higher Y-BOCS total and subscale scores, HAMD, and HAMA scores relative to HCs (Table 1).

### 3.2. Group Differences of fALFF within the Fronto-Limbic Network 

Compared with the HCs, OCD had increased fALFF value in the right OFC (Table 2 and Figure 1).

### 3.3. Group Deviations of NH within the Fronto-Limbic Network

OCD displayed increased NH in the left OFC and decreased NH in the right putamen compared with HCs (Table 2 and Figure 2).

### 3.4. Relationship between fALFF/NH Values and Clinical Characteristics in OCD

The decreased NH value of the right putamen was negatively related to the Y-BOCS total scores (*r* = −0.332, *p* = 0.036) and compulsive behavior scores (*r* = −0.336, *p* = 0.034) (Figure 3).

### 3.5. SVM Results

SVM analysis proceeded with three abnormal fALFF/NH values (1 = right OFC, 2 = left OFC, 3 = right putamen) discovered in OCD and pairwise combinations. The results were as follows: 1 accuracy = 76.92% (60/78; classification); 2 accuracy = 78.21% (61/78; classification); 3 accuracy = 70.51% (55/78; classification); 12 accuracy = 80.77% (63/78; classification); 13 accuracy = 79.49% (62/78; classification); and 23 accuracy = 89.74% (70/78; classification). The accuracy of the combination of 2 and 3 was the highest (Figure 4), and thus it could be utilized to distinguish OCD from HCs with a sensitivity of 95.00% (38/40), a specificity of 84.21% (32/38), and an accuracy of 89.74% (70/78) (Figure 5).

## 4. Discussion

In the current research, fALFF and NH approaches were used to investigate the regional- and network-level brain activities within the fronto-limbic network at rest in medicine-free OCD. The results manifested that increased fALFF in the right OFC, increased NH in the left OFC, and decreased NH in the right putamen were discovered at rest in OCD. In addition, decreased NH of the right putamen was negatively related to the Y-BOCS total and compulsive behavior scores. A combination of NH in the left OFC and right putamen could be utilized to identify OCD from HCs with optimum specificity and sensitivity.

Consistent with previous neuroimaging studies, our findings manifested increased fALFF value in the right OFC at rest in OCD [27,28,29]. Moreover, we discovered the increased NH value in the left OFC in OCD. As a component of the fronto-limbic network, the OFC has a crucial role in OCD etiology by emotional regulation and reward processing [30]. Increased fALFF and NH in the OFC may underpin aberrant coordination within the fronto-limbic network at rest, and it may be related to more effort to modulate negative emotion (i.e., anxiety and fear) and tolerance of uncertainty in patients with OCD. By contrast, the changed brain regions were not the same using fALFF and NH approaches in the current study. We infer that the increased spontaneous neuronal activity of the right OFC may contribute to enhanced connectivity of the left OFC within the fronto-limbic network at rest due to the dynamic brain networks [25].

Our research revealed decreased NH of the right putamen at rest in patients with OCD. Previous studies found increased gray-matter volume, lowered functional connectivity at rest, and increased activation during emotional processing in the right putamen in OCD [2,31,32]. As a part of the striatum, putamen receives and integrates information from the cortex [33]. Furthermore, putamen, a key region of the fronto-limbic network, participates in the sensorimotor network in OCD [6]. OCD pathology (i.e., maladaptive habituation behavior mediated by the sensorimotor network) may cause the decreased NH of the right putamen at rest in OCD [6], which may interpret the relationship between decreased NH values of the right putamen and clinical symptoms of OCD discovered in the current study. The SVM results showed that a combination of NH in the left OFC and right putamen may be used to differentiate individuals with OCD from HCs, suggesting the crucial role of the network-level brain activities of the fronto-limbic network at rest in the neurocircuit-based classification in patients with OCD.

In the fronto-limbic model, we found that patients with OCD showed increased regional- and network-level brain activities in the frontal cortex and decreased network-level in the limbic system at rest. Notably, these brain regions also participate in other networks (i.e., ventral affective and sensorimotor network) involving other functions [6]. Therefore, the fronto-limbic network may work with other networks involved in OCD [6].

As important brain regions of the fronto-limbic network, vmPFC, ACC, and amygdala showed no altered regional- and network-level brain activities at rest in patients with OCD in our research, which was inconsistent with previous findings [14,15]. Heterogeneity of clinical samples (i.e., sample sizes, comorbidity, and medication status) may explain these inconsistencies [34]. In addition, the low statistical power of a relatively small sample size may lead to the low reproducibility of neuroimaging results [35]. The activities of these brain regions are highly sensitive to emotional stimuli and symptomatic provocation but not particularly changed at rest [1].

Compared with medicine-free patients, OCD-medicated patients showed decreased activation of the OFC during symptom provocation tasks [36] and increased functional connectivity between the right putamen and the left frontal cortex at rest [37]. Drug treatment may normalize OCD-related impaired segregation of the functional network in the whole brain [38]. For this reason, some patients with OCD had a history of psychotropic medication in the current study, which may affect the local and network properties of the fronto-limbic network at rest. Some possible unmeasured variables, such as social effects and physical effects, may also affect the current results [39,40]. For these reasons, the current findings should be prudently interpreted.

This study has several limitations. First, we explored the regional- and network-level brain activities in OCD at rest, not the task state, which is closely related to the function of the fronto-limbic network. Second, we did not investigate the causality between the cortical and subcortical brain areas of the fronto-limbic network in OCD. Finally, whether the current altered regional- and network-level brain activities change with treatment needs to be determined in longitudinal studies.

## 5. Conclusions

We found altered regional- and network-level brain activities within the fronto-limbic network in medicine-free and non-comorbidity patients with OCD at rest. Our findings emphasize the crucial role of the fronto-limbic network in the etiology of OCD.

## Figures and Tables

**Figure 1 brainsci-12-00857-f001:**
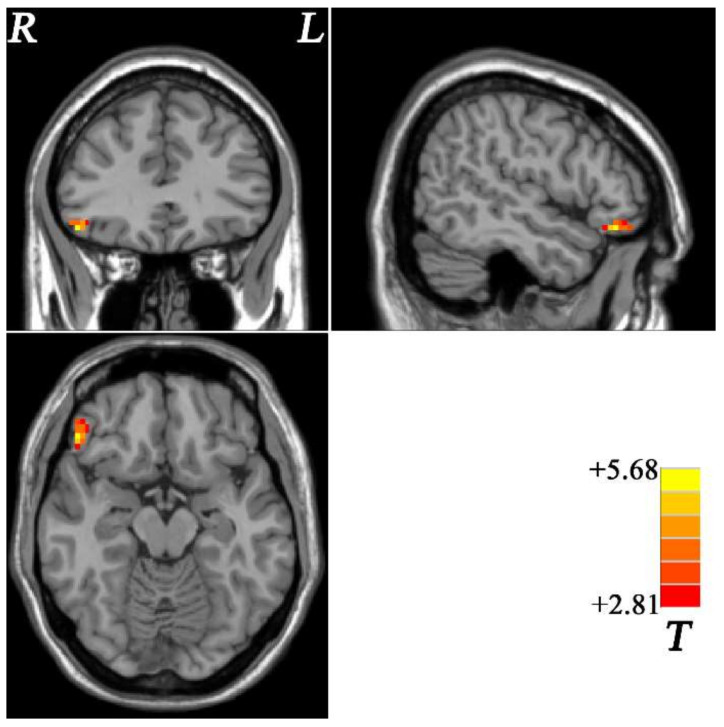
Brain regions with altered fALFF within fronto-limbic network at rest in OCD. The color bar indicates the *T* values from two-sample *t*-tests. fALFF = fractional amplitude of low-frequency fluctuations, OCD = obsessive–compulsive disorder, L = left, R = right.

**Figure 2 brainsci-12-00857-f002:**
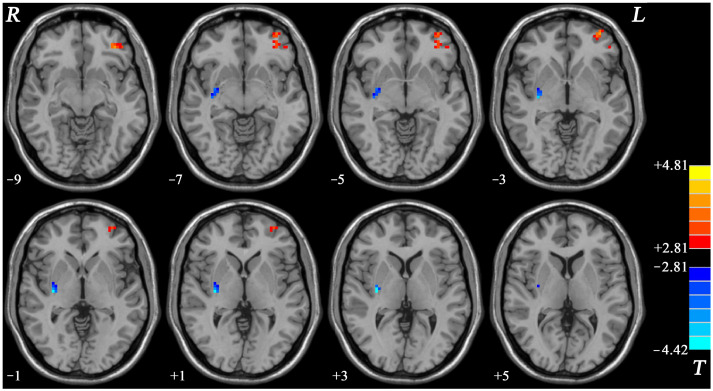
Brain regions with changed NH of fronto-limbic network at rest in OCD. The color bar indicates the *T* values from two-sample *t*-tests. NH = network homogeneity, OCD = obsessive–compulsive disorder, L = left, R = right.

**Figure 3 brainsci-12-00857-f003:**
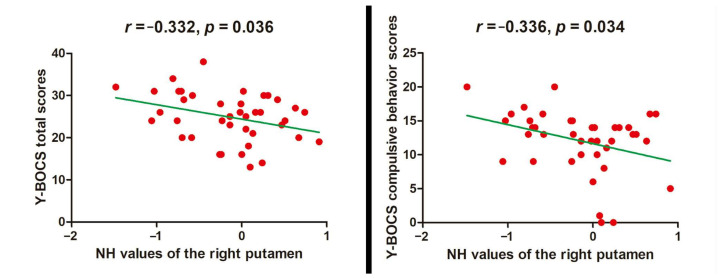
Relationship between NH values and clinical characteristics in OCD. NH = network homogeneity, OCD = obsessive–compulsive disorder, Y-BOCS = Yale–Brown Obsessive–compulsive Scale.

**Figure 4 brainsci-12-00857-f004:**
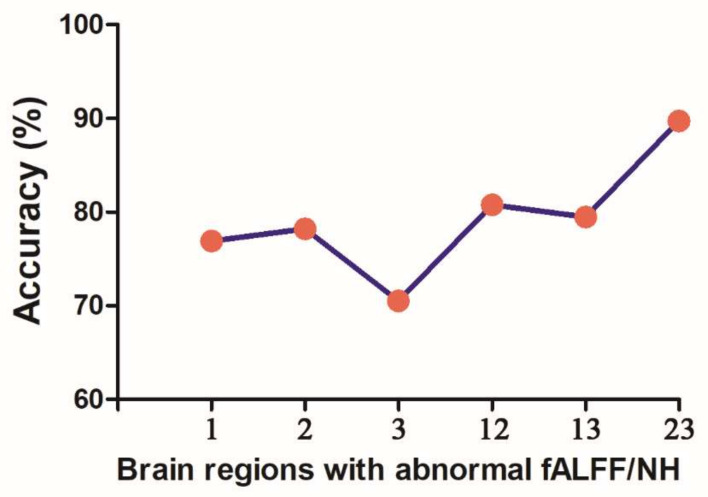
Accuracy (%) of SVM using three brain regions with altered fALFF/NH values of fronto-limbic network to discriminate OCD from HCs. fALFF = fractional amplitude of low-frequency fluctuations, NH = network homogeneity, 1 = right orbitofrontal cortex, 2 = left orbitofrontal cortex, 3 = right putamen, SVM = support vector machine, OCD = obsessive–compulsive disorder, HCs = healthy controls.

**Figure 5 brainsci-12-00857-f005:**
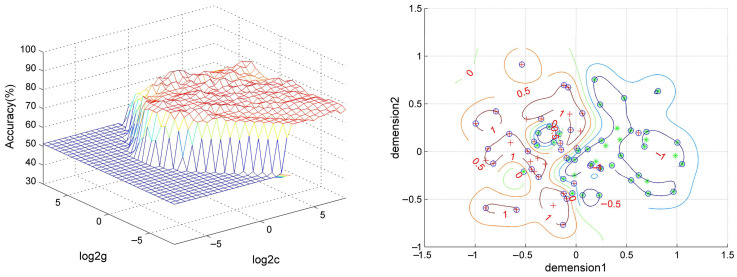
Visualization of SVM results using NH values of left orbitofrontal cortex and right putamen. Left: 3D visualization of SVM with the best parameters; right: classification map of the NH values of left orbitofrontal cortex and right putamen. SVM = support vector machine, NH = network homogeneity, log 2c and log 2g = the range and step size of c and g (c and g are the parameters of the kernel functions).

**Table 1 brainsci-12-00857-t001:** Sociodemographic and clinical characteristics of participants.

	OCD Patients (n =40)	HCs (n = 38)	*X^2^/t*	*p*
Age (years)	27.28 ± 8.16	27.18 ± 8.33	0.05	0.71
Sex (male/female)	27/13	25/13	0.03	0.87
Education (years)	13.40 ± 2.87	13.74 ± 3.03	−0.50	0.83
Illness duration (months)	66.68 ± 75.54			
Y-BOCS total score	24.90 ± 5.73	1.13 ± 0.88	25.27	<0.01
Y-BOCS obsessive thinking	12.85 ± 4.25	0.37 ± 0.49	17.98	<0.01
Y-BOCS compulsive behavior	12.05 ± 4.62	0.74 ± 0.72	14.92	<0.01
HAMD	8.05 ± 4.40	1.45 ± 0.95	9.04	<0.01
HAMA	10.83 ± 6.55	1.16 ± 1.00	9.00	<0.01
FD	0.04 ± 0.02	0.03 ± 0.01	1.25	0.13

OCD = obsessive–compulsive disorder, Y-BOCS = Yale–Brown Obsessive–Compulsive Scale, HAMD = 17-item Hamilton Depression Rating Scale, HAMA = Hamilton Anxiety Rating Scale, FD = framewise displacement. Variables of age, education, Y-BOCS total score, subscales score, HAMD score, HAMA score, and FD were tested by two-sample t-tests, the results were indicated by t values. Categorical data such as gender were tested using a chi-squared test; the result was indicated by X^2^.

**Table 2 brainsci-12-00857-t002:** Regions with altered fALFF/NH in fronto-limbic network at rest in OCD.

Cluster Location	Peak (MNI)			Number of Voxels	*T* Value
	x	y	z		
**fALFF**					
Right OFC	51	33	−15	30	5.6825
**NH**					
left OFC	−36	42	−9	50	4.0674
Right putamen	33	−12	3	25	−4.4232

*p* < 0.05 corrected by Gaussian Random Field (GRF) (voxel significance: *p* < 0.001, cluster significance: *p* < 0.05). fALFF = fractional amplitude of low-frequency fluctuations, NH = network homogeneity, OCD = obsessive–compulsive disorder, OFC = orbitofrontal cortex, MNI = Montreal Neurological Institute.

## Data Availability

Our data may be available upon reasonable request. Please contact lipingchxyy@163.com for details.

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
