# Peer review of "Altered Regional Activity and Network Homogeneity within the Fronto-Limbic Network at Rest in Medicine-Free Obsessive–Compulsive Disorder"

_brainsci, 2022, doi:10.3390/brainsci12070857_

Round 1
Reviewer 1 Report
In my opionion patients' sample data should be better described. Maybe a table should be added with socio-demographic and clinical characteristics. Data of interest for your research is also duration of illness.
Some information about therapy should be added: patients that were treated was treated only with SSRIs? how long did therapy last? low doses of SGAs were added?
Author Response
1.In my opionion patients' sample data should be better described. Maybe a table should be added with socio-demographic and clinical characteristics. Data of interest for your research is also duration of illness.
Response: Thanks for the reviewer’s suggestion. We have added Table 1 with Socio-demographic and clinical characteristics of participants in the revised manuscript.
2. Some information about therapy should be added: patients that were treated was treated only with SSRIs? how long did therapy last? low doses of SGAs were added?
Response: Eighteen patients with OCD were drug-naive; fourteen patients had a history of anti-obsessive/antidepressant/ anxiolytic medication (i.e., selective serotonin reuptake inhibitors, serotonin and norepinephrine reuptake inhibitors, benzodiazepines, and buspirone); and eight patients had a history of antipsychotics (i.e., aripiprazole 5-20 mg/day).

Reviewer 2 Report
-What percentage of patients were undergoing behavioral therapy? Was this an exclusion or how was this accounted for if allowed?
-Was OCD the only psychiatric diagnosis?
-were patients free of other physical co-morbidities?
-minor, first line of the results seems to have a symbol typo on the statistics for gender.
-what are some of the results in this brain region for OCD-medicated patients? Can you compare and contrast in the discussion?
-the consideration of possible unmeasured variables having a confounding effect on your results should be considered in the discussion (social effects, other physical effects like weight, etc).
Author Response
1.What percentage of patients were undergoing behavioral therapy? Was this an exclusion or how was this accounted for if allowed?
Response: None of the patients were undergoing any systematic cognitive behavioral therapy or behavioral therapy.
2. Was OCD the only psychiatric diagnosis?
Response: YES. Please see the exclusion criteria of all participants in the Materials and Methods Participants.
3. were patients free of other physical co-morbidities?
Response: YES. Please see the exclusion criteria of all participants in the Materials and Methods Participants.
4. minor, first line of the results seems to have a symbol typo on the statistics for gender.
Response: Thanks for the reviewer’s suggestion. We have corrected in the revised manuscript.
5. what are some of the results in this brain region for OCD-medicated patients? Can you compare and contrast in the discussion?
Response: Compared with medicine-free patients, OCD-medicated patients showed decreased activation of the OFC during symptom provocation tasks [36] and increased functional connectivity between the right putamen and the left frontal cortex at rest [37]. Drug treatment may normalize OCD-related impaired segregation of functional network in the whole-brain [38]. For this reason, some patients with OCD had a history of psychotropic medication in the current study, which may affect the local and network properties of the fronto-limbic network at rest. Please see Discussion in the revised manuscript.
6. the consideration of possible unmeasured variables having a confounding effect on your results should be considered in the discussion (social effects, other physical effects like weight, etc).
Response: Thanks for the reviewer’s suggestion. Some possible unmeasured variables, such as social effects and physical effects, may also affect the current results [39-40]. For these reasons, the current findings should be prudently interpreted. Please see Discussion in the revised manuscript.

Round 2
Reviewer 2 Report
I have no further comments
Author Response
Dear Professor,
Thanks a lot for your hard work.